# Evaluation of Atrial Electromechanical Delay in Children with Obesity

**DOI:** 10.3390/medicina55060228

**Published:** 2019-05-30

**Authors:** Fatih Temiz, Hatice Güneş, Hakan Güneş

**Affiliations:** 1Department of Pediatric Endocrinology, Sutcu Imam University, 46100 Kahramanmaras, Turkey; ftemiz2003@yahoo.com; 2Department of Pediatrics, Sutcu Imam University, 46040 Kahramanmaras, Turkey; 3Department of Cardiology, Sutcu Imam University, 46040 Kahramanmaras, Turkey; drhakangunes83@gmail.com

**Keywords:** atrial electromechanical delay, children, obesity

## Abstract

*Background and Objective*: Childhood obesity is one of the worldwide health problems with an increasing prevalence and accompanied by severe morbidity and mortality. It is a serious predisposing risk factor especially for the development of cardiovascular diseases and arrhythmias. Electromechanical delay (EMD) is known to be a predictor for the development of atrial fibrillation (AF). Our study aims to investigate whether EMD, which is a predictor of AF, prolongs in obese children or not. *Material and Methods*: The study included 59 obese patients aged between 8–18 years and 38 healthy patients as the control group with a similar age and gender. All the individuals underwent transthoracic echo and tissue Doppler echocardiography. Systolic and diastolic left ventricular (LV) functions, inter- and intra-atrial electromechanical delay were measured by tissue Doppler imaging (TDI) and conventional echocardiography. *Results*: Obese patients had significantly lengthened P-wave on surface ECG to the beginning of the late diastolic wave (PA) lateral, PA septum, intra- and inter-atrial electromechanical delays when compared with the control group (*p <* 0.001, *p* = 0.001, *p* < 0.001 and *p* < 0.001, respectively) Inter-atrial EMD and intra-atrial EMD correlated positively with body mass index (BMI) values (*r* = 0.484, *p* < 0.001 and *r* = 0.376, *p* = 0.001; respectively) BMI was significantly related with inter-atrial EMD (*β* = 0.473, *p* < 0.001) However, there was no relationship between inter-atrial EMD and serum glucose and platelet count. *Conclusion*: In our study, we declared that electromechanical delay was increased in obese children when compared to the control group and intra- and inter-atrial electromechanical delay was in correlation with body mass index. Furthermore, we discovered that BMI is an independent predictor of the inter-atrial EMD in obese children.

## 1. Introduction

Childhood obesity is one of the most important worldwide public health problems [1,2]. Overweight and obesity prevalence is increasing in both developed and developing countries, in some regions the rate of obesity is one-third of the adolescent population. This increased prevalence has led to an increase in the obesity-related co-morbid diseases [1,3]. Obesity in early life is considered as a risk factor for cardiovascular morbidity and mortality [4,5,6,7,8,9,10,11,12].

As well as obesity at an early age is closely related to ischemic heart disease, it is also related to non-ischemic heart diseases such as arrhythmias; obesity increases the risk of supra-ventricular arrhythmia such as atrial fibrillation [13,14]. Childhood obesity is a major risk factor for atrial fibrillation whereas structural remodeling is very important [15].

P wave dispersion (PWD) is a very useful non-invasive method for the use of surface electrocardiography as it shows the heterogeneity of repolarization in atrial myocardium. PWD has been evaluated in many childhood and adult diseases. However, there are few studies on PWD in childhood obesity [16,17,18,19]. In addition to electrocardiography (ECG), non-invasive echocardiography markers are clinically useful for the prediction of arrhythmia that occurs in various diseases. EMD is one of them and is defined as the time between the occurrence of force in the myocardium and the onset of electrical activity. The EMD, which is easily obtainable by tissue Doppler imaging (TDI), also indicates atrial conduction heterogeneity [20]. It is well known that increased atrial EMD in TDI is an indicator of supra-ventricular tachycardia development [21].

To the best of our knowledge, TDI has not been used for the detection of an atrial electromechanical delay in obese child subjects. In this study, we examined whether atrial EMD detected by TDI was prolonged in obese child subjects or not. In addition, we investigated the relationship between PWD with EMR in obese children.

## 2. Materials and Method

### 2.1. Study Population

This prospective cross-sectional study included 59 obese children aged 8–18 years, who were admitted to the pediatric endocrinology and metabolism outpatient clinic of Kahramanmaraş Sütçü imam University between July 2018 and December 2018. As the control group, 38 volunteers and healthy individuals with a similar age and gender who applied to the outpatient clinic were included in the study. Patients’ anthropometric measurements such as height and weight were performed only with underwear and without shoes. BMI (kg/m^2^) was calculated by dividing a person’s weight in kilograms by their height in meters squared.

Obesity was defined as the BMI index being above the 95th percentile according to specific percentile curves determined with age and gender. An individual was considered as morbidly obese if they were in the 99th percentile [22].

All patients underwent 12-lead electrocardiography (ECG). Inclusion criteria were determined as follows: Volunteering to participate in the study, obese and 8–18 years of age, ECG with a sinus rhythm. Exclusion criteria were as follows: Patients with known congenital heart disease, patients with branch block in the ECG, patients with atrial septal defect, patients with anti-arrhythmic drug use for any reason (such as beta blocker, digoxin), patients with rheumatic valve disease, active infection, known diabetes mellitus (DM), Cushing syndrome, patients with hypertension, chronic inflammatory disease, hypothyroidism or hyperthyroidism and patients who had previously been diagnosed with supra-ventricular tachycardia. Laboratory and demographic data of the subjects were recorded. Standard echocardiography and tissue Doppler echocardiography were performed on all the patients by the same cardiologist during the outpatient visit.

### 2.2. Electrocardiography (ECG) and P Wave Dispersion (PWD) Measurement

After 10 min of rest, 12 lead ECGs were taken at a 50 mm/s rate and 20 mm/mV amplitude in the supine position. These ECGs were evaluated by two different cardiologists who were blinded of the clinical condition of the patients. The beginning of the P wave was determined as the first point where the first deviation of the P wave intercepted the isoelectric line and the end as the end point of the end of the deviation. PWD was measured and defined as the variation of the difference between maximum P wave (Pmax) and the minimum P wave (Pmin) time calculated from leads D2 and V5.

### 2.3. Standard Echocardiography

Transthoracic echocardiographic examinations were performed by experienced echocardiographers who were blinded about the clinical information of the subjects via, the Vivid 7^®^ cardiac ultrasonography system (GE Ving-Med Ultrasound AS; Horten, Norway) with 2.5- to 5-MHz probes. The supine and left lateral positions were conducted for each patient with 2D, M-mode, pulsed and color flow Doppler echocardiography. Single lead electrocardiogram continuously recorded. For all measurements, the average of at least three cardiac cycles was evaluated. For the conventional Doppler echocardiographic examinations and M-mode measurements, the European Society of Echocardiography guideline criteria were used [23]. Two-dimensional images and doppler tracings were obtained from parasternal short and long axes, subcostal and apical views. Right, and left atrial dimensions, left ventricular (LV) end-systolic and end-diastolic dimensions and diastolic LV septal and posterior wall thickness were measured. While the disc method was used to measure left atrial volumes, the Simpson rule was used for predicted left ventricular ejection fraction (EF). Mitral inflow velocities, peak E (early diastolic) and peak A (late diastolic), E/A ratio were obtained for the evaluation of the LV diastolic function, and also deceleration time of the E-wave (DT) and isovolemic relaxation time (IVRT) was used.

### 2.4. Tissue Doppler Echocardiography (TDE) 

The pulsed Doppler sample volume was placed at the LV lateral mitral, RV tricuspid and septal mitral annulus in an apical four-chamber view. The time was defined as PA and was obtained from the lateral mitral annulus (PA lateral), septal mitral annulus (PA septal) and RV tricuspid annulus (PA tricuspid) from the starting of the P wave in the surface ECG to the beginning of the late diastolic wave (Am). While intra-atrial EMD was PA septum minus PA tricuspid, the inter-atrial EMD was calculated and defined as PA lateral minus PA tricuspid [24,25] (Figure 1).

### 2.5. Ethics Statement

This study was approved from Kahramanmaraş Sütçüimam University Ethics Committee with the protocol code 258 on 13 July 2018. Informed consent was provided before starting the study from all individuals.

### 2.6. Statistical Analysis

Data management and analysis was performed by using the SPSS program v.14 (SPSS Inc., Chicago, IL, USA) and a two-sided *p*-value ≤0.05 was considered as statistically significant. Categorical variables were determined by the percentage and number of cases, while continuous variables were expressed as the mean ± standard deviation (SD) or median and interquartile range (IQR). A mean was compared by using an independent sample *t*-test, and in the case of an abnormal distribution Mann–Whitney U test with the median was used. A chi-square test was used for the categorical data. Correlation analyses of abnormally distributed variables were performed by Spearman correlation analysis and Pearson correlation was used in the normal distributed variables. A stepwise multiple regression analysis was used for the definition of the significant determinants of inter-atrial EMD, and incorporating variables that correlated with a *p*-value of less than 0.1 in the correlation analysis. A value of *p* < 0.05 was considered as statistically significant.

## 3. Results

Table 1 represents the clinical, laboratory and echocardiographic findings of the groups. Age, gender and heart rate were similar between the two groups, whereas BMI was significantly higher in obese patients (*p* < 0.001). When laboratory characteristics of the two groups were compared, serum glucose and platelet counts (*p* = 0.016 vs. 0.022) were significantly higher in the obese group, while the other values were similar between the two groups. LV ejection fraction, LA diameter, septum thickness and posterior wall thickness, A velocity, E velocity and E/A ratio were similar between the two groups (*p >* 0.05; Table 1).

Data are given as mean ± standard deviation (SD) number and percentage, *p* ≤ 0.05 was considered statistically significant.

Compared to the healthy controls, PA septum, PA lateral, inter- and intra-atrial electromechanical delays were prolonged in the obese group (*p* = 0.001, *p* < 0.001, *p* < 0.001 and *p* < 0.001, respectively; Table 2).

Inter-atrial EMD and intra-atrial EMD correlated positively with BMI values (*r* = 0.484, *p* < 0.001 and *r* = 0.376, *p* = 0.001, respectively; Figure 2 and Figure 3). Stepwise linear regression analysis demonstrated that BMI was significantly related with inter-atrial EMD (*β* = 0.473, *p* < 0.001) but, there was no relationship between serum glucose level and platelet count with inter-atrial EMD.

Obese patients had significantly lengthened Pmax, Pmin and PWD compared with healthy control subjects (*p <* 0.001, *p* = 0.007, *p* < 0.001 and *p* < 0.001, respectively; Table 3). 

P wave dispersion correlated positively with inter-atrial EMD and intra-atrial EMD values (*r* = 0.737, *p* < 0.001 and *r* = 0.382, *p* < 0.001, respectively; Figure 4 and Figure 5).

## 4. Discussion

To our knowledge, this study is the first study investigating the electromechanical delay in obese children, and we showed that electromechanical delay was increased in obese children compared to the control group. Additionally, we found that intra- and inter-atrial EMD’s correlate with BMI. Furthermore, we concluded that BMI is an independent predictor of inter-atrial electromechanical delay in obese child patients.

Blood glucose levels are high in obese patients; it is thought that this is due to the insulin resistance that accompanies obesity [1]. High blood glucose levels trigger neutrophil release from the S100 calcium binding protein (S 100A8/A9), which binds to advanced glycation and end products (RAGE) receptors in liver Kuppfer cells. This release of neutrophils leads to an increase in thrombopoietin (TPO) levels. TPO causes proliferation of megakaryocytes and increased platelet production. Parallel with the literature, in our study, fasting blood glucose levels, and platelet counts were found to be high in obese patients [26,27].

Obesity is closely associated with metabolic diseases, coronary heart disease, heart failure and especially with arrhythmias such as atrial fibrillation (AF) in childhood, as it is in the adult age group [28,29,30]. The incidence of atrial fibrillation, which is common in the adult age group and which causes increased mortality and morbidity, has also begun to increase in the childhood period. Childhood AF usually occurs in patients with valvular and congenital heart disease due to the increased stress of the left atrium [31]. Recent studies in pediatric patients have begun to identify patients with atrial fibrillation that occur in the absence of an underlying heart or systemic disease and proved that children might be affected by these arrhythmias. El-Assaad et al. found that obesity is one of the critical risk factors for atrial fibrillation [15]. The study by Frost et al. showed that high body fat was related with AF development [29]. Schmidt et al. have shown the risk of developing AF of overweight and obese young men was twice the risk of healthy young people [32].

Although the mechanism of obesity is not fully understood, it is suggested that AF develops due to mechanisms such as fat inflammation on the atrial wall, the increase in sympathetic nervous system activity, increased inflammatory process, adipokinin dysregulation and activation of pro-fibrotic signaling pathways [30,33,34]. These complex mechanisms in the atrium wall cause an extension in the electromechanical conduction time. In the study of Erdem et al., the EMD time was higher in the adult obese patients compared to the control group [35]. Similar findings were found in the study of Yagmur et al. [36]. In our study, similar to these studies, we found that the duration of the intra- and inter-electromechanical delay was longer in obese children. The previous studies showed that prolonged atrial electromechanical conduction time predisposed the development of supra-ventricular arrhythmias, particularly atrial fibrillation [23].

PWD, being an ECG index for reflecting heterogeneous atrial conduction through detecting abnormal atrial conduction with ECG leads to a different orientation. It has been studied for several times on patients with hypertension, metabolic syndrome, DM and obesity as a simple and noninvasive predictor of AF development [17,18,19]. Although the accurate mechanism of PWD prolongation in obesity is not a well-known issue, it is believed to have structural and electrophysiological changes in the atrial myocardium because of fatty inflammation caused by obesity. Fatty inflammation can cause a number of disorders by making chemical changes in the structure and function of proteins in the cell membrane [34]. Furthermore, in obese subjects the autonomic control of the heart is abnormal because of the prevalence of sympathetic over parasympathetic limb of the autonomic balance. As a result, the autonomic disparity noticed in obese subjects may have an effect on intra-atrial and inter-atrial conduction times, and leave them prone to develop atrial arrhythmias such as atrial fibrillation [19]. Uner et al. reported that PWD was significantly higher in obese patients [37]. Similarly, in this study, we found that Pmax, Pmin and PWD were significantly higher in obese patients and there was an important correlation between inter-atrial and intra-atrial EMD and PWD. In light of this information, the delay in electromechanical conduction time and PDW may be related to atrial fibrillation in obese children.

Our study also showed that BMI, which was closely associated with left atrial fibrosis and left atrial enlargement, was positively correlated with intra- and inter-atrial conduction, and BMI was found to be an independent predictor of inter-atrial electromechanical delay.

### Study Limitations

Since the design of the study was cross-sectional, patients could not be followed up for long-term AF development. Additionally, the variability in inter-observer PA measurement was not evaluated. Inter-atrial conduction time obtained by TDI was not correlated with invasive inter-atrial conduction time. Lastly, in order to support our hypothesis, there is a need for studies including a large number of subjects and long-term follow-up.

## 5. Conclusions

In conclusion, the delay of the electromechanical conduction time in obese patients may be due to the atrial inflammatory and fibrotic process associated with obesity. Elongation in the electromechanical period may be the early finding for atrial fibrillation that developed or will be developed in obese children. Close follow-up of these patients may help to prevent the early complications.

## Figures and Tables

**Figure 1 medicina-55-00228-f001:**
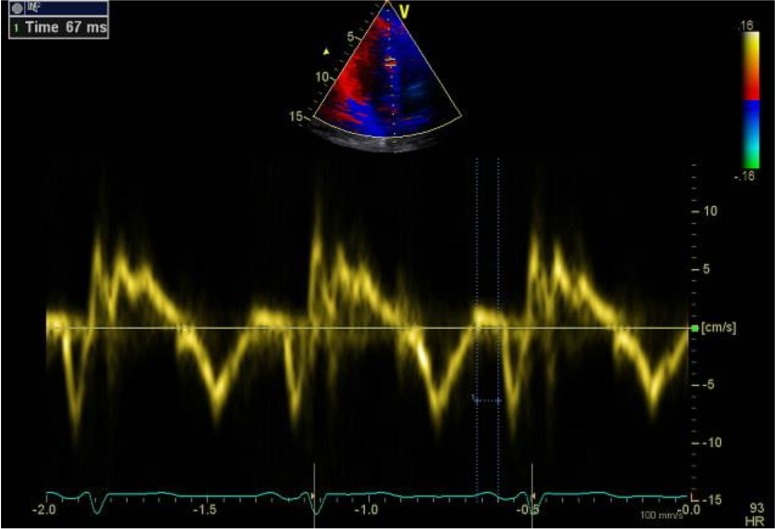
Measurement of the time interval from the onset of the P wave on the surface electrocardiography (ECG) to the starting of late diastolic wave (Am wave) interval with tissue Doppler imaging (PA tricuspid, PA septal and PA lateral, respectively).

**Figure 2 medicina-55-00228-f002:**
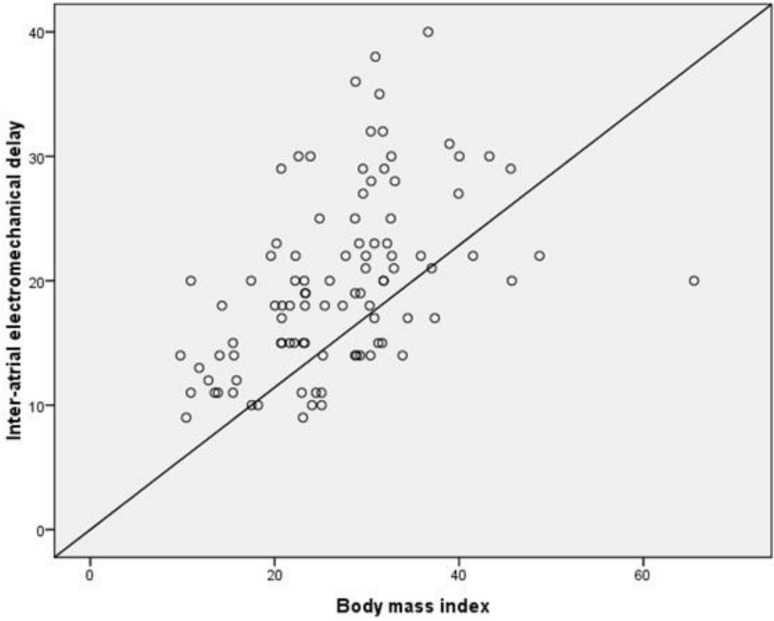
Correlation between inter-atrial electromechanical delay (EMD) and body mass index.

**Figure 3 medicina-55-00228-f003:**
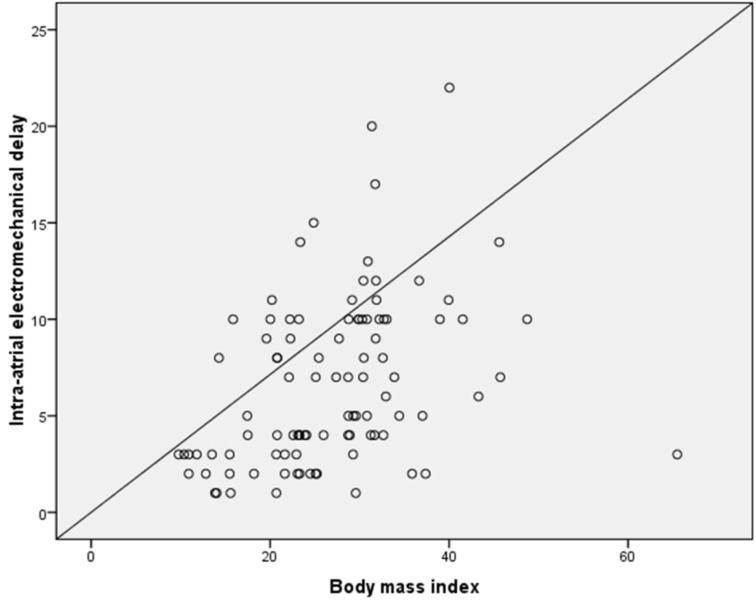
Correlation between intra-atrial EMD and body mass index.

**Figure 4 medicina-55-00228-f004:**
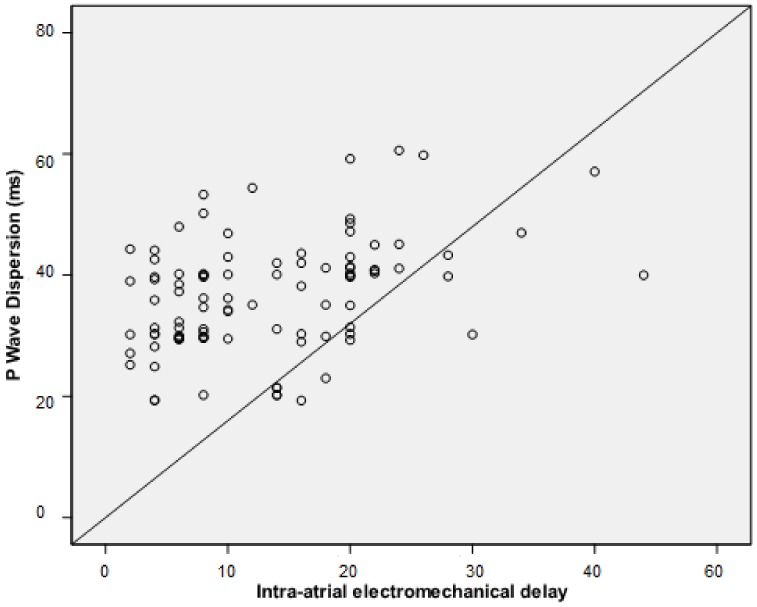
Correlation between intra-atrial EMD and P wave dispersion.

**Figure 5 medicina-55-00228-f005:**
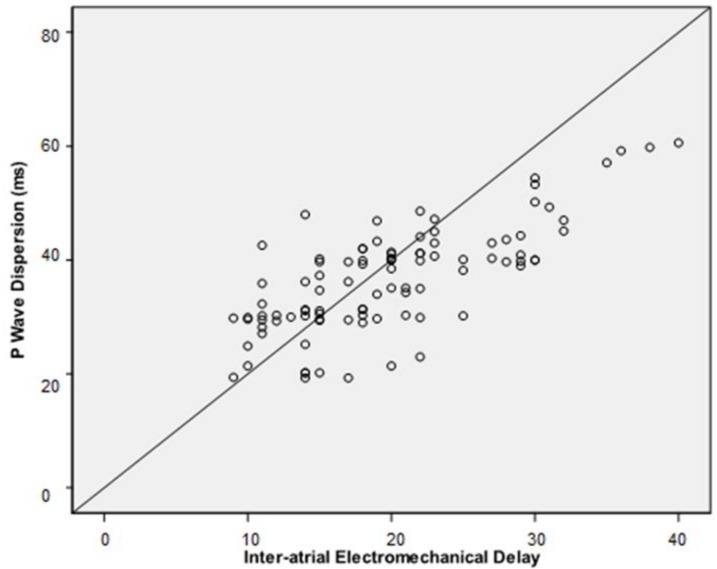
Correlation between inter-atrial EMD and P wave dispersion.

**Table 1 medicina-55-00228-t001:** Baseline characteristics of the study patients.

	Obese Subjects (*n* = 59)	Non-Obese Subjects (*n* = 38)	*p*
Age, mean ± SD, years	13.4 ± 2,7	13.4 ± 2.9	0.963
Male/female, *n*	22/37	7/31	0.079
Body mass index, mean ± SD	31.5 ± 8.5	19.6 ± 4.3	<0.001
Heart rate, mean ± SD, beats/min	73 ± 8	72 ± 6	0.132
Blood glucose, mean ± SD, mg/dL	88 ± 8	84 ± 8	0.016
ALT, median (IQR), U/L	22 (17–33)	16 (12–23)	0.154
AST, median (IQR), U/L	24 (20–28)	24 (20–30)	0.114
Urea, mean ± SD, mg/dL	10.1 ± 2.3	10 ± 2.1	0.772
Total protein, mean ± SD, g/dL	7.3 ± 1.1	7.0 ± 0.7	0.160
Hemoglobin, mean ± SD, g/dL	13.6 ± 1.1	13.3 ± 1.5	0.364
Platelet count, mean ± SD, 10^3^/mm^3^	333 ± 84	292 ± 83	0.022
RDW, mean ± SD, %	14.1 ± 3.3	13.5 ± 1.1	0.288
LV ejection fraction, mean ± SD, %	70 ± 2	70 ± 2	0.883
IVS, mean ± SD, mm	8.0 ± 0.2	7.2 ± 0.2	0.132
Posterior wall thickness, mean ± SD, mm	7.9 ± 0.2	7.4 ± 0.2	0.232
Mitral E velocity, mean ± SD, cm/s	95 ± 14	94 ± 14	0.924
Mitral A velocity, mean ± SD, cm/s	66 ± 17	63 ± 17	0.440
E/A, mean ± SD	1.5 ± 0.3	1.4 ± 0.3	0.187
Left atrial diameter, mean ± SD, cm	3.1 ± 0.4	3.0 ± 0.3	0.516

ALT: Alanine aminotransferase, AST: Aspartate aminotransferase, RDW: Red cell distribution width, LV: Left ventricle, and IVS: Interventricular septum thickness. Data are given as mean ± standard deviation (SD) number and percentage, or median and interquartile range (IQR). *p* ≤ 0.05 was noted statistically significant.

**Table 2 medicina-55-00228-t002:** Comparison of the atrial electromechanical coupling parameters measured by tissue Doppler imaging.

	Obese Subjects (*n* = 59)	Non-Obese Subjects (*n* = 38)	*p*
PA Septum, mean ± SD, ms	45 ± 6	41 ± 5	0.001
PA Lateral, mean ± SD, ms	60 ± 8	52 ± 7	<0.001
PA Tricuspid, mean ± SD, ms	37 ± 5	36 ± 5	0.299
Inter-atrial EMD, mean ± SD, ms	22 ± 7	16 ± 5	<0.001
Intra-atrial EMD, median (IQR), ms	7 (4–10)	4 (2–8)	<0.001

PA = The time interval from the beginning of the P wave in the surface ECG to the beginning of the Am wave interval in tissue Doppler echocardiography, and EMD = electromechanical delay. Data are presented as mean ± standard deviation (SD) number and percentage, or median and interquartile range (IQR). *p* ≤ 0.05 was considered statistically significant.

**Table 3 medicina-55-00228-t003:** Comparison of obese and control groups with regard to the maximum P wave (Pmax), minimum P wave (Pmin) and P wave dispersion (PWD).

	Obese Subjects (*n* = 59)	Non-Obese Subjects (*n* = 38)	*p*
Pmax, median (IQR), ms	98.1 (89.3–104)	85.2 (76.4–98.1)	<0.001
Pmin, median (IQR), ms	57.3 (55–60)	49 (45.2–64.3)	0.007
Pd, median (IQR), ms	40 (34–43.6)	30.3 (29.3–39.4)	<0.001

PWD, P dispersion; Pmax, maximum P wave duration and Pmin, minimum P wave duration. Data are presented as median and interquartile range (IQR). *p* ≤ 0.05 was considered statistically significant.

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
