# Peer review of "Evaluation of Atrial Electromechanical Delay in Children with Obesity"

_medicina, 2019, doi:10.3390/medicina55060228_

Round 1

Reviewer 1 Report

The authors present a single centre experience in Echocardiography in obese chidren.

The paper is surely well written studies but it has important shortcomings that the autors should fill.

You could increase the number of the studied population in order to create more BMI interval categories. Furthermore, as you noticed in the limitation section, you haven't perform a comparison of variability among different operators in echo measurements; I think that it shold be mandatory and easily achieved.

Finally, the electromechanical delay is a predictor of arrhythmais but you did't correlate this datum to any "electrophysiological measure". I think that, if is not possible to perform an electrophysiological study, neverthless some electrophysiological aparameters should be acquired (i.e  P wave duration, supraventricular arrhythmias at ECG Holter recording, HRV, etc)

In conclusion, I think the manuscript should be strengthen with the acquisition if new data and results.

Author Response

Response to Reviewer 1 Comments

Point 1: You could increase the number of the studied population in order to create more BMI interval categories. Furthermore, as you noticed in the limitation section, you haven't perform a comparison of variability among different operators in echo measurements; I think that it shold be mandatory and easily achieved.

Response 1: Thanks to your valued contributions, P dispersion was evaluated by two different cardiologists. EMD measurements have been done to 10 children by cardiologists who participated and did not participate in the study to Show variance only among operators. Similar results were obtained in both measurements. As this procedure was not conducted during the research, it was not reported in the study.

Point 2: Finally, the electromechanical delay is a predictor of arrhythmais but you did't correlate this datum to any "electrophysiological measure". I think that, if is not possible to perform an electrophysiological study, neverthless some electrophysiological aparameters should be acquired (i.e  P wave duration, supraventricular arrhythmias at ECG Holter recording, HRV, etc)

Response 2: We tried to support our study with the findings of ECG (Pmax, Pmin, and p dispersion) since we could not perform Holter ECG follow-up and electrophysiological study (an invasive procedure) because ethical approval was not obtained due to patients being asymptomatic. The relationship between P dispersion and EMD was shown by graphs.

Reviewer 2 Report

Temiz and colleagues present a manuscript on atrial electromechanical delay in obese children. The study design is cross-sectional and uses matched controls. The methods are well chosen and described, the results are clearly depicted.

However, I have the following concerns and comments:

-          Unfortunately, the study does not include any electrical follow-up data (i.e. Holter recordings showing frequency of premature beats, episodes of tachycardia etc.) to support the link between an early increase in EMD and arrhythmogenicity.

-          The focus of the introduction lies on general arrhythmogenicity, while later, the focus is shifted to AF. The focus should be coherent throughout the manuscript.  

Minor:

-          line 39: sinus arrhythmia is not a disease

-          line 57-58: sentence should be rephrased, in the current word order, the nurse is the one only wearing underwear and shoes

-          line 67: should state beta blocker

-          figure 1: looks like a photograph of an echo print image, digitally exported image should be used

Author Response

Response to Reviewer 2 Comments

Point 1: Unfortunately, the study does not include any electrical follow-up data (i.e. Holter recordings showing frequency of premature beats, episodes of tachycardia etc.) to support the link between an early increase in EMD and arrhythmogenicity.  The focus of the introduction lies on general arrhythmogenicity, while later, the focus is shifted to AF. The focus should be coherent throughout the manuscrip.  

Response 1: We attempt to provide compatibility in the article by correcting the shift between the general arrhythmia and AF in the introduction. Since the study did not include parameters that require follow-up such as Holter records, it was attempted to support the study with p wave dispersion in the ECG drawn to the patients.

Point 2: Minor:

-          line 39: sinus arrhythmia is not a disease

-          line 57-58: sentence should be rephrased, in the current word order, the nurse is the one only wearing underwear and shoes

-          line 67: should state beta blocker

-          figure 1: looks like a photograph of an echo print image, digitally exported image should be used

Response 2:  The proposed changes are highlighted in the text. Digitally exported echo image is taken and added to the text.

Round 2

Reviewer 1 Report

no further comments